# Risk Management and Recommendations for the Prevention of Fatal Foreign Body Aspiration: Four Cases Aged 1.5 to 3 Years and Mini-Review of the Literature

**DOI:** 10.3390/ijerph17134700

**Published:** 2020-06-30

**Authors:** Angelo Montana, Monica Salerno, Alessandro Feola, Alessio Asmundo, Nunzio Di Nunno, Filomena Casella, Emilpaolo Manno, Federica Colosimo, Raffaele Serra, Giulio Di Mizio

**Affiliations:** 1Department of Medical, Surgical and Advanced Technologies G.F. Ingrassia, University of Catania, Via S. Sofia 87 Edificio B, 95123 Catania, Italy; monica.salerno@unifg.it; 2Department of Experimental Medicine, University of Campania “Luigi Vanvitelli”, Via Luciano Armanni 5, 80138 Naples, Italy; alex.feola@gmail.com; 3Department of Clinical and Experimental Medicine—Dermatology, University of Messina, “G. Martino” Hospital, 98124 Messina, Italy; aasmundo@unime.it; 4Department of History Society and Human Studies, University of Salento-Lecce, 73100 Lecce, Italy; nunzio.dinunno@icloud.com; 5Forensic Medicine, Department of Law, Magna Grecia, University of Catanzaro, 81100 Catanzaro, Italy; mena.casella84@gmail.com (F.C.); federicacolosimo@virgilio.it (F.C.); giulio.dimizio@unicz.it (G.D.M.); 6Emergency Department, Maria Vittoria Hospital-Torino, 10144 Torino, Italy; manno@aslto2.it; 7Department of Surgical and Medical Sciences, Magna Graecia University of Catanzaro, 81100 Catanzaro, Italy; rserra@unicz.it

**Keywords:** foreign body aspiration, health promotion, autopsy, children, prevention, community

## Abstract

(1) Background: Foreign body aspiration (FBA) is a significant public health concern among the pediatric population, and fatalities are dramatic for families. It typically involves organic foreign bodies (mainly food) aspirated by children under three years old, usually at home or school. This review aimed to focus on the preventive measures around four actual cases of fatal foreign body aspiration, emphasizing the correct execution of the Heimlich maneuver and cardiopulmonary resuscitation, supervised mealtimes, and high-risk foods. (2) Methods: Four fatal cases of foreign body aspiration in children are presented here. The children were in a free environment, such as school, home, and the countryside, and were in the presence of teachers, parents, and a grandmother who did not supervise the children adequately. A literature review was performed via the MEDLINE database using the key terms: “foreign body aspiration,” “infant choking, 1.5 to 3 years,” “food and foreign body aspiration,” “common household,” “prevention of foreign body aspiration,” “guidelines,” “recommendations,” “training of caregivers (parents, educators),” “resuscitation,” “Heimlich maneuver,” and “disengagement of the upper airways.” We focused on the prevention of foreign body aspiration. (3) Results: a complete postmortem examination was performed. In three cases, the foreign bodies were food (mozzarella cheese, pear, or raw bean), while in one case, the foreign body was a pebble. (4) Conclusions: This review aimed to discuss recent scientific literature and provide a perspective on the benefits of a dedicated approach to the management of fatal foreign body aspiration in children by caregivers who usually have no experience with the best ways of supervising children in a safe environment, especially regarding the correct execution of resuscitation maneuvers, such as the Heimlich maneuver. Recommendation updates could improve healthcare quality in a pediatric setting and reduce medico-legal implications.

## 1. Introduction

Foreign body aspiration (FBA) is a life-threatening pediatric emergency due to the accidental impaction of objects in the respiratory tract and represents a significant public health problem, sometimes resulting in fatal outcomes [1,2]. The injuries, which are causes of morbidity and mortality in all age groups, are seen mainly in children under 3 years old and is the fourth leading cause of accidental death in this group and the third in infants under 1 year [3].

The extra-hospital mortality rate is about 36.4%. The in-hospital mortality rate for airway foreign bodies is between 0.26% and 13.6% after complications due to delayed hypoxia (severe laryngeal edema or bronchospasm requiring a tracheotomy or re-intubation, pneumothorax, pneumomediastinum, cardiac arrest, tracheal or bronchial laceration, and hypoxic brain damage) [3,4,5,6,7].

Food foreign bodies are the most common items identified in choking events, where boys less than five years of age may be at the highest risk [8]. The lack of private insurance has also been associated as a risk factor for foreign body aspiration in children; this may be related to a potential lack of anticipatory guidance and caretaker education. 

It may be important to understand the utilization of resources and identify whether there is a need to formulate targeted educational awareness programs, anticipatory guidance for at-risk pediatric patient populations, or more cost-effective management strategies [9]. Inpatient hospitalizations for children account for a significant economic burden and airway foreign bodies are an example of a clinical condition in children that is usually associated with inpatient hospitalization. 

Cheng et al. demonstrated that the charges associated with airway foreign bodies in children appear to be rising as well. We have found that there is a clear and increasing trend in the cost of treatment for children with airway foreign bodies [10].

FBA has a higher incidence between 1 and 3 years of age, and in children under 1 year old, it is the most likely cause of accidental fatalities [11,12,13]. A foreign body in the respiratory tract is one of the most common causes of accidental death at home in the <3-year-old age group, accounting for 7% of sudden deaths in children up to 3 years old. In the United States, FBA causes 5% of all accidental deaths in children under the age of 4 years old and is the leading cause of accidental deaths in the home among children under the age of six [14]. The lack of molar teeth; poor swallowing of food; playing with objects in the mouth; their tendency to put objects in the mouth; talking, crying, or moving while eating; and having weak protective laryngeal reflexes are indicated as primary risk factors in younger children [15,16]. The Susy Safe register [17], one of the largest international registers that collect cases of foreign body aspiration accidents (corresponding to the codes of the “International Classification of Diseases, Ninth Revision,” Clinical Modification from 930 to 939) in children aged between 0 and 14 years [18], shows that the foods that most often cause accidents are chicken and fish bones (32%), followed by peanuts (22%) and seeds (16%) [19].

However, the foods that most often cause these types of accidents are not the same ones that cause the most severe consequences: in the Susy Safe register, serious accidents are caused by meat, while a study conducted in 26 Canadian and US hospitals showed how sausages are more often associated with fatal episodes. These observations are linked to the fact that the severity of the outcome depends on specific characteristics of the shape, consistency, and size of the food [20]. A foreign body made of inorganic material can go undetected for a long time, whereas an organic foreign body can give origin to an inflammatory process that can quickly provoke obstructive symptoms. A large foreign body can cause complete obstruction of the airway and rapid death. Complete airway obstruction is most likely to occur with round or egg-shaped foreign bodies that adapt to the airway shape of the child [7]. The Susy Safe register shows that almost half of the foreign body accidents occurred under adult supervision [21]. These results indicate a lack of knowledge from adults responsible for child supervision regarding the problem of FBA, as confirmed by a recent survey conducted on families with children under four years old [22].

Increased awareness of the parents, teachers, caregivers, and health providers could play an essential role in lowering the risk of this potentially life-threatening situation.

This review aimed to emphasize the importance of education about choking hazards and how to reduce the incidence of these events. Pediatricians play a crucial role in promoting injury prevention too. Mortality can be reduced by educating caregivers and relatives about potentially fatal risks. The aim is to highlight specific recommendations related to the correct execution of resuscitation maneuvers, such as the Heimlich maneuver, which is essential for children’s survival.

## 2. Materials and Methods

### 2.1. Literature Review

We used MEDLINE to search the English, German, Spanish, and French language literature from 1995 to 2020 for articles using the following search terms: “foreign body aspiration,” “infant choking, 1.5 to 3 years,” “food and foreign body aspiration,” “common household,” “prevention of foreign body aspiration,” “guidelines,” “recommendations,” “training of caregivers (parents, educators),” “resuscitation,” “Heimlich maneuver,” and “disengagement of the upper airways.” We focused on the prevention of foreign body aspiration. All sources were screened independently by three of the authors to determine their relevance in the framework of the current report and were selected for inclusion by two of the authors.

### 2.2. Case Reports

Written informed consent was obtained from the parents of the children for the publication of this case report.

#### 2.2.1. Case 1

A 3-year-old girl who ate mozzarella cheese in her kindergarten dining room, sitting at a table, had a silent cough, was unable to vocalize, unable to breathe, became cyanotic, and had a decreasing level of consciousness. The kindergarten teacher reported that the girl had had a frequent cough during the day.

The teachers tried to give her first aid through several back blows with no success and contacted the emergency medical services. By the time the emergency medical personnel arrived on the scene, the child was no longer there, having been transported to the closest hospital by car by one of the teachers. In the emergency room, after inspection of the upper airway, a trained anesthesiologist attempted endotracheal intubation with the use of a laryngoscope multiple times, with no success. The patient arrived at the hospital emergency department in about 10 min after the initial incident and the physicians declared death after 20 min of resuscitation maneuvers for foreign body ingestion asphyxia. No evidence of a foreign body in the airway was reported in the medical record of the anesthesiologist. External examination of the body presented congestion and edema of the face, as well as petechial hemorrhages in the skin of the face and the lining of the eyelids. The autopsy showed massive edema glottidis and a wedge-shaped piece of mozzarella cheese, 5 cm long and 2 cm wide, obstructing the aditus ad laringem. The lung parenchyma appeared to be increased in volume and the consistency and result from palpation were emphysematous. Histological examination showed acute pulmonary emphysema and a mild alveolar hemorrhage. Peribronchiolar lymphocytes and neutrophils were detected in the lungs and laryngeal mucosa, while a toxicological examination of urine and blood samples excluded the presence of drugs. These data confirmed that the little girl had had acute bronchitis and this could have facilitated the inhalation of food during a coughing spell. The cause of death was mechanical asphyxiation caused by the inhalation of food facilitated by coughing and bronchitis.

#### 2.2.2. Case 2

A 2-year-old boy was playing on the lap of his grandmother, who was sitting at the kitchen table and cleaning some raw beans. Suddenly, the child picked up one of the beans and put it in his mouth. The child showed a severe cough, biphasic stridor, dry cough (tracheal type) with a sharp crack, and subsequent loss of consciousness. The grandmother tried unsuccessfully to remove the foreign body with her fingers. Emergency services arrived on the scene in a short time (20 min), but they were unable to resuscitate the child, who was then pronounced dead.

External examination of the body presented slight congestion of the face, as well as petechial hemorrhages in the skin of the face and the lining of the eyelids. The autopsy revealed the presence of a raw bean obstructing the middle third of the trachea with a diameter of 1 cm; the bean measured 3 × 1.5 cm.

The lung parenchyma appeared increased in volume and the consistency was emphysematous during palpation. Inspection of the lung and heart parenchyma highlighted small sub-pleural and sub-epicardial petechiae. Acute pulmonary emphysema and edema were observed at the histological examination. Toxicological examination of urine and blood samples excluded the presence of substances of abuse. Mechanical asphyxiation due to acute obstruction of the upper airways was indicated as the cause of death.

#### 2.2.3. Case 3

An 18-month-old boy was playing with his parents in the countryside, eating a large piece of pear. The child’s parents, who were away for a few seconds, reported that the child suddenly developed a cough, wheezing, decreased breathing sounds, and cyanosis. The parents immediately called the emergency medical services, who sent an air ambulance, followed by a land ambulance, when the helicopter was unable to locate the scene of the incident due to its remote and unclear location. By the time the land ambulance arrived on the scene, about 30 min after the initial emergency call, the child had died. The body presented stab wounds on the right side of the neck as a result of the desperate efforts by the father, who had no medical training, to perform an emergency tracheotomy. The parents reported that they tried mouth-to-mouth resuscitation with no success. External examination of the body identified slight cyanosis (blue discoloration) of the skin and cyanosis of the face; an incision in the lower neck through the skin but not in the airway (trachea) was observed. The autopsy revealed slight edema glottidis and a whole fragment of pear, measuring 2.5 × 1.5 cm, obstructing the trachea at the bifurcation (diameter of 1 cm). Toxicological examination of urine and blood samples excluded the presence of drugs. Acute pulmonary emphysema and mild pulmonary edema were observed at the histological examination. Death was caused by acute respiratory insufficiency due to mechanical, violent, accidental asphyxia due to a bolus obstruction (fruit).

#### 2.2.4. Case 4

A 3-year-old boy was playing alone on the home balcony when he put a pebble that was in a vase in his mouth. The father did not pay attention due to undisclosed other activities inside the house but hearing a severe cough, he went out onto the balcony to help the boy. The child developed a wheeze, stridor, breathlessness, vomiting, and cyanosis too. The father realized that a pebble was blocking the larynx, and he tried to remove it with fingers but without success and contacted the emergency medical services. Emergency services arrived on the scene after 15 min, transported the child to an emergency room with subsequent admission to intensive care. The physician attempted endotracheal intubation with the use of a laryngoscope multiple times, with no success. For the first two days, chest X-rays and chest tomographies (TCs) were negative for the presence of the pebble. Only on the third day did the physicians detect the pebble, which was then removed with a laryngoscope; the diameter of the pebble was 3 cm. The child died after six days in a coma from hypoxia resulting from respiratory arrest, as well as anoxic brain injury. External examination of the body identified subconjunctival and facial petechiae. Organs examined during the autopsy showed evident signs of diffused visceral congestion. The lung parenchyma appeared increased in volume and the consistency was emphysematous during palpation. The trachea had a diameter of 1.5 cm. Acute pulmonary emphysema and edema were observed under histological examination.

## 3. Results

In the four cases presented here, in contrast to the “European Resuscitation Council Guidelines for Resuscitation 2010, Section 6. Pediatric life support” [23], resuscitative maneuvers were unsuccessful or were not performed at all by caregivers (parents, grandparents, and teachers). A summary of the details of the four cases is reported in Table 1.

In case 1, a 3-year-old ate mozzarella cheese in her kindergarten dining room, sitting at a table. The child developed signs of airway obstruction and the teachers tried to give her aid through several back blows with no success. Unfortunately, the Italian school system does not make it mandatory for teachers to attend first aid classes [24]. These guidelines report that in the absence of legislation in this regard, it is strongly recommended to provide for the constant presence of personnel who have followed a first aid course in the school canteens frequented mainly by children. Furthermore, the same guidelines recommend that families and adults responsible for supervising the child (e.g., educators in kindergartens, summer centers, after-school, teachers, babysitters) know the rules regarding food preparation and behavior at the table for the prevention of suffocation from food (for example, in the case of the use of plastic cutlery, ensure that these are hard and resistant); furthermore, families and adults responsible for the child’s supervision should acquire knowledge and skills regarding maneuvers to unblock airways and cardiopulmonary resuscitation. In this case, the Heimlich maneuver and adequate supervision of the child were needed. The child arrived at the hospital emergency department where the physicians declared death from suspected asphyxiation, which was later confirmed by autopsy.

In case 2, the child (a 2-year-old) was playing on the lap of his grandmother, who was sitting at a table cleaning some raw beans when the child picked up one of the beans and put it in his mouth. The grandmother tried unsuccessfully to remove the foreign body with her fingers. In this case, the Heimlich maneuver would have been crucial, but unfortunately, the grandmother did not know what to do after the foreign body was consumed by a child that shows the symptoms of airway obstruction.

In case 3, the child (an 18-month-old) was playing with his parents in the countryside, eating a large piece of pear. While the parents were away for a few seconds, the child showed symptoms of airway obstruction. The child died at the investigation scene before any emergency services arrived. In this case, the parents did not perform the Heimlich maneuver and tried to perform an inaccurate tracheotomy that in this case would not have been recommended; moreover, it is essential to avoid offering high-risk foods, such as a large piece of pear, since 18-month-old children can safely eat pear if it is cut into very small pieces [25].

In case 4, the child (a 3-year-old) was playing alone on the balcony of his home when he put a pebble that was in a vase in his mouth; he then developed symptoms of airway obstruction. The father was not paying attention to the child due to undisclosed other activities inside the house. He tried to remove the object with his fingers but without success. In this case, the father should have supervised the child on the balcony at home; kept hazardous objects out of reach, such as the pebbles; and the Heimlich maneuver was necessary.

## 4. Discussion

To prevent injury, the supervision of children is essential in all aspects of their lives, even during eating. Most choking episodes occur during meals or play and generally occur under adult supervision [26,27,28,29].

In the four cases analyzed, the supervision of the children by caregivers was inadequate or completely absent. In fact, in case 1, the child consumed food brought from home, and the teachers did not supervise the child adequately during lunch and were not able to perform the Heimlich maneuver, only giving her aid through several back blows with no success. In case 2, the child was playing on the lap of his grandmother who was sitting at a table cleaning some raw beans when the child picked one up and put it in his mouth. In this case, it was partially correct to attempt to remove the foreign body with fingers but the grandmother did not supervise the child that was on her lap and did not perform the Heimlich maneuver; beans, because of their shape, are considered high-risk foods. In case 3, the child was playing with his parents and ate a large piece of pear while the parents were away for a few seconds and the child showed symptoms of airway obstruction. In this case, the parents should have supervised the child and not left the child for a few seconds. In case 4, the child picked up a pebble and put it in his mouth; the father did not pay attention because of undisclosed other activities inside the house. In this case, the father should have supervised the child on the balcony at home and kept hazardous objects, such as pebbles, out of reach. In this regard, to prevent infant choking, the Mayo Clinic [7,30,31] suggests avoiding these three incorrect behaviors: (1) introducing the baby to solid foods before they have the motor skills to swallow them can lead to infant choking; (2) giving babies or young children hot dogs, chunks of meat or cheese, grapes, raw vegetables, or fruit chunks, unless they are cut up into small pieces; (3) allowing the child to play, walk, or run while eating. The aspiration of a foreign body is an event that is reasonably frequent and very dramatic in children and is one of the leading causes of mortality and morbidity.

The majority of foreign body aspirations occur below the age of three years (76.9%). In our sample, along with other reviews, a male predominance was observed [32,33,34,35].

Foreign body aspirations can be attributed to the tendency of children to explore their world via the mouth, incomplete development of the full posterior dentition, and their immature neuromuscular mechanism for swallowing and airway protection [36,37,38,39,40,41,42,43]. The high predominance of boys in these studies (75%) was related to their higher activity, as demonstrated in other accident statistics [22,23,44,45,46,47]. The data collected by the Susy Safe register show that food is responsible for up to 26% of injuries due to the insertion/aspiration/inhalation/ingestion of FBs. Traditionally, these types of incidents have been regarded as unavoidable accidents, and only recently have they been recognized as being eligible for preventive efforts [48,49,50]. Notably, children should eat appropriate types of food according to their age. As reported by Wu et al., food, other than milk, may be introduced into the infant’s diet but should be done carefully [51].

Solid food should never be given before the fourth month and it is better not to provide solid food until after the seventh or eighth month when the teeth and salivary glands have begun to develop. Between the tenth and twelfth months, breastfeeding may be gradually suspended but milk should still form the staple feed up to the age of eighteen months. At this age, a little meat may be wisely introduced in the solid form to provide some practice in masticating [52].

One of the risk factors for FBA is a lack of knowledge by caregivers of children, where the European-Union-based surveillance of FBA and the Susy Safe registry show that an adult was present in 40% of the FBA cases involving a child younger than 1 year old [13,14,15,16,17,18,19,20,21,22,23,24,25,26,27,28,29,30,31,32,33,34,35,36,37,38,39,40,41,42,43,44,45,46,47,48,49,50,51,52,53].

The prevention of choking on food by young children should include public education regarding surveillance and training of cardiopulmonary resuscitation maneuvers, cautionary food labeling, recalls when necessary, and actions to encourage food manufacturers to provide further attention to child safety and modify their products to prevent choking-related injuries. Caregivers must be confident with the best ways of supervising children in a safe environment. However, when precautions fail and aspiration is suspected or witnessed, parents should perform first aid procedures and call the emergency services and ask for help if necessary. Meanwhile, basic life support maneuvers should be started (such as the Heimlich maneuver, which is considered life-saving in cases of foreign-body airway obstruction) as soon as possible [15,33,54]. Therefore, this review highlights the importance of the Heimlich maneuver, which in the four cases reported here, was not performed by any adult present (teachers, grandmother, parents, and father for each case, respectively). Abdominal thrusts or the Heimlich maneuver is a first-aid procedure used to treat upper airway obstruction caused by a foreign body [55].

Despite the rare instances of intra-abdominal complications, the Heimlich maneuver is considered a quick and inexpensive technique that does not require medical knowledge or expertise to perform [56,57,58,59]. Today, the Heimlich maneuver is accepted and taught during Basic Life Support (BLS) and Advanced Cardiovascular Life Support (ACLS) for conscious adults but backslaps are still recommended for infants and chest compressions are recommended for unconscious patients. A child with a presumed airway obstruction that is still able to maintain some degree of ventilation should be allowed to clear the airway by coughing. If the child cannot cough, vocalize, or breathe, emergency steps are necessary to clear the airway. For infants under one year of age, alternating sequences of five back blows and five chest thrusts are performed until the object clears or the infant becomes unresponsive. Abdominal thrusts should not be performed on infants, as their livers are more prone to injury.

For a choking child over one year of age, subdiaphragmatic abdominal thrusts (i.e., the Heimlich maneuver) should be performed until the object is cleared or the child becomes unconscious [60,61,62,63]. If the infant or child becomes unresponsive, immediately start chest compressions. After 30 compressions, the airway should undergo evaluation. If a foreign body is visible, it requires removal, but blind finger sweeps should not be performed as they may push the foreign body downwards to the larynx [64,65,66]. A series of 30 compressions and two breaths should continue until the object is expelled [67,68,69,70]. The “European Resuscitation Council Guidelines for Resuscitation 2010, Section 6” [71,72] on pediatric life support recommends an algorithm for managing foreign body airway obstruction (FBAO) [60,73,74,75] (see Figure 1).

## 5. Conclusions

A critical strategy to decrease the risk of foreign body aspiration and to prevent infant choking includes the need to inform the public (caregivers, parents, families) about the importance of performing the Heimlich maneuver [76,77]. Equally important strategies involving properly timing the introduction of solid foods, not offering high-risk foods, and supervising mealtimes. Increased public awareness and the extensive use of the Heimlich maneuver could diminish the mortality from acute obstruction. The most critical factor in reducing mortality is the recognition of a person with acute airway obstruction [3]. Complete foreign body airway obstruction is a medical emergency and requires immediate action by untrained bystanders to restore the victim’s airway. The American Academy of Pediatrics (AAP) recommends choking first aid and CPR training for parents, teachers, childcare providers, and others who care for children [78]. The Heimlich maneuver exemplifies what can be achieved by involving the general population in community healthcare [79]. The overall improvement in patient outcomes has been found to rely heavily on bystander cardiopulmonary resuscitation and basic life support [80]. Therefore, since education plays a crucial role in injury prevention, providing counseling about safe behaviors should be included in all visits to pediatricians to make parents conscious of the risks associated with eating some foods and enable them to select a safe environment for their children. Sensitizing the community toward the need for awareness regarding FBA through educational campaigns involving stricter mass media measures can drastically reduce the morbidity and mortality associated with FBA. Healthcare professionals and public health experts should create a national monitoring center for research and surveillance on this phenomenon; family and hospital pediatricians and physicians must also be committed during consultancy activities to raising families’ awareness of the risk of suffocation and to guide them in making the most appropriate food choices for their children’s health.

## Figures and Tables

**Figure 1 ijerph-17-04700-f001:**
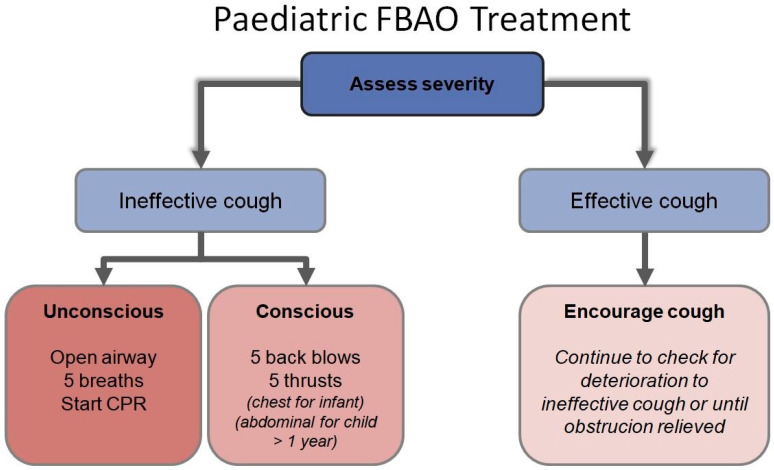
Pediatric foreign body airway obstruction (FBAO) algorithm. CPR: Cardiopulmonary resuscitation.

**Table 1 ijerph-17-04700-t001:** Data regarding the deceased.

Cases	Age/Sex	Foreign Body	Location	Place of the Event	Caregivers	Mistakes in Resuscitation	Outcome
1	3 y.o./F	Mozzarella cheese (5 × 2 cm)	Aditus ad laringem	Kindergarten; sitting at a table	Teachers	The teachers did not supervise the child adequately during lunch and were not able to perform the Heimlich maneuver; they only performed several back blows with no success	The child died in the hospital emergency department
2	2 y.o./M	Raw bean (3 × 1.5 cm)	Middle third of the trachea	Home; child was playing on the lap of his grandmother who was sitting at a table and cleaning some raw beans in the kitchen.	Grandmother	In this case, it was necessary to perform the Heimlich maneuver and supervise mealtimes to stop the child from playing, walking, or running while eating	The child died at home; emergency services arrived on the scene in a short time, but they were unable to resuscitate
3	18 months/M	Large pear fragment (2.5 × 1.5 cm)	Trachea at the bifurcation	Countryside; child was playing with his parents, ate a large piece of pear while the parents were away for a few seconds, and showed symptoms of airway obstruction	Parents	Parents did not perform the Heimlich maneuver; father did not perform an accurate tracheotomy, which in this case was not recommended; high-risk foods, such as a large piece of pear, should not be offered	The child died at the investigation scene before emergency services arrived
4	3 y.o./M	Pebble (diameter of 3 cm)	Aditus da laringem	Home; child was playing alone on the balcony when the father, who did not pay attention due to undisclosed other activities inside the house, heard a severe cough and went out onto the balcony to help the child	Father	Avoid introducing the fingers into the child’s mouth; keep hazardous objects out of reach (common household items that might pose a choking hazard include coins, button batteries, dice, pen caps, vases with pebbles); take a class on cardiopulmonary resuscitation (CPR) and choking first aid for children; perform the Heimlich maneuver	The child died in hospital after being in a coma for 6 days

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
