# Peer review of "Risk Management and Recommendations for the Prevention of Fatal Foreign Body Aspiration: Four Cases Aged 1.5 to 3 Years and Mini-Review of the Literature"

_ijerph, 2020, doi:10.3390/ijerph17134700_

Round 1
Reviewer 1 Report
This manuscript offers a nice summary of prevention of foreign body aspiration in young children, as well as four engaging case studies that illustrate the risks and prevention strategies. I have several comments for the authors to consider.
- I don’t understand the purpose of the literature review. What did it offer? What was the goal? Was it systematic or just thorough? How were the results of the literature review used to accomplish the aims of the study? I feel like the current citations are appropriate and probably the result of the literature review, but the details of how the literature review was conducted may not be needed, or else should be clarified to explain what the goals and process were.
- In the case of the grandmother and the child who ate a raw bean (case 2), I question whether supervision was inadequate here. From what is presented, it seems the child was well supervised – after all, the child was sitting on his grandmother’s lap! To me, the problem in this case was lack of knowledge about the risks, and then lack of knowledge about what to do after the foreign body was consumed. So I agree mistakes were made, but I don’t feel inadequate supervision was a factor in this particular case based on the information provided.
- Lines 47-49 – could the authors provide more details on the rates of fatality or injury? Present data to demonstrate the scope or significance of the problem more clearly.
- Lines 50-55 – this paragraph confuses me. I’m not sure how it fits the paper and it seems out of place. It might possibly be deleted.
- In case 1, it seems we must require teachers of young children to learn and know the risks and the prevention strategies. Policies are needed at the local, state/province, or national level. This might be incorporated into the study recommendations more clearly?
- Line 193-194 – related to point #2 above, I don’t think the boy was playing, walking, or running in this case. The risks were related to lack of knowledge on the part of the supervisor.
- Line 199-200 – is the problem offering high-risk foods, or is the problem not cutting the food into much smaller pieces? I believe medical experts would suggest 18-month-old children can safely eat pear if it is cut into very small pieces, is that correct?
- The manuscript is a bit repetitive – in many cases, the same information is presented in the case reports, the results section, and again in the table. Streamlining is suggested.
- Line 236 – the 2.7:1 female:male ratio is presented here for the first time and surprises me. This seems a high difference and might be something to discuss, especially since 3 of the 4 cases presented were male, not female.
- Line 240-241 – I’m not sure I agree that these types of incidents were regarded as unavoidable accidents, at least not by injury scholars over the last 25 years or more. Perhaps this sentence should be clarified, revised, or include a citation?
- Lines 257-258 – Is calling emergency services the best option? In some of the cases, the delay in response created a more serious medical situation. It seems the better prevention strategy is for parents and supervisors to be trained in prevention (both primary and secondary), including the Heimlich maneuver.
Reviewer 2 Report
Had better add the quality of literature evaluation; do not limit the year of study and collect as much literature as possible.
